# The Role of Gender and Age in the Emotional Well-Being Outcomes of Young Adults

**DOI:** 10.3390/ijerph18020522

**Published:** 2021-01-10

**Authors:** Claudia López-Madrigal, Jesús de la Fuente, Javier García-Manglano, José Manuel Martínez-Vicente, Francisco Javier Peralta-Sánchez, Jorge Amate-Romera

**Affiliations:** 1School of Education and Psychology, University of Navarra, 31009 Pamplona, Spain; jdlfuente@unav.es; 2Institute for Culture and Society, University of Navarra, 31009 Pamplona, Spain; jgmanglano@unav.es; 3School of Psychology, University of Almería, 04120 Almería, Spain; jvicente@ual.es (J.M.M.-V.); fjperalta@ual.es (F.J.P.-S.); jorgeamate@yahoo.es (J.A.-R.)

**Keywords:** young adults, coping strategies, self-regulation, resilience, positivity

## Abstract

Young adults face different stressors in their transition to college. Negative emotions such as stress can emerge from the demands they face. This study aimed at gaining an improved understanding of the role that gender and age play in the well-being of young adults. Coping strategies, resilience, self-regulation, and positivity were selected as indicators of well-being. Descriptive and inferential analysis have been conducted. Results show that well-being varies significantly with age and gender. Gender was predominantly involved in the acquisition of the well-being outcomes, highly predicting problem-focused coping strategies. No interaction effects were found between gender and age. An improved understanding of the developmental factors involved in well-being outcomes will enlighten future interventions aimed at improving young people’s resources to face adversity.

## 1. Introduction

Each developmental period has its own sources of stress and is associated with various social contexts, which lead to differences in the use of psychosocial strategies [1]. Young adults (18–25 years old) are subject to a number of stressors that affect their social, emotional, and relational spheres. Uncertainty about the future, academic pressure, and social integration are stressors that influence their performance [2,3,4]. A growing number of recent studies seek to evaluate ways of promoting well-being and positive adaptation among youth. Resilience, positivity, self-regulation, and coping strategies are considered protective factors among college students, shielding them from negative health outcomes [5,6]. However, most of the available literature is focused on children and adolescents. More evidence is needed on how these variables are affected by other sociodemographic variables over time, such as gender and age.

### 1.1. Young Adulthood as a Critical Developmental Stage

In western industrialized societies, sociocultural change has made the path to adulthood longer, diverse, and complex [7]. The peculiarities of this stage have led to define it as a separate period in the life-course, emerging adulthood, from ages of 18 to 25 years old, approximately [4]. Although there is still debate on the term, and whether it can be considered as a new developmental stage, there is an agreement that the traditional model of transition from adolescence straight to adulthood has changed. Beyond infancy, there is no other life-course stage that “experiences such dynamic and complex changes on the personal, social, emotional, neuroanatomical, and developmental levels” [8].

Five main domains characterize this stage [9]. First, it is an age of identity explorations, especially in areas such as love and work, in which youth learn more about themselves and what they want in life. Second, constant shifting in love and work makes this period one of constant instability and self-exploration. Third, it is a self-focused stage, in which people detach from their parents and start looking for their own standards of self-sufficiency, progress, and achievement. Fourth, it is an age of feeling “in-between” meaning that they perceive themselves neither as adolescents nor full adults. The constant exploration of new possibilities and the instability of the transition itself triggers this ambiguous feeling. Lastly, this has been defined as an age of unlimited possibilities and great expectations, leading new opportunities for flourishing.

From a neurodevelopmental perspective, these years are a time of risk and opportunity. Brain development continues developing after adolescence. Neocortical association areas and the frontal lobes continue maturing until the mid-twenties [10]. The brain starts to change in response to the environment and prepares for the challenges of adulthood [11]. Adult functions are developed through important changes in the limbic and frontal systems involved in attention, avoidance, and engagement, as well as in other social processes [12]. The amount of white matter increases, while gray matter decreases [13]. At this age, the development of the brain plays an important role. Changes in neurocognitive skills are influenced by social and psychological experiences; functional connections of the brain are essential for the development of psychological functions [14]. The development of the frontal circuits after adolescence are influenced by events, such as full-time employment, advanced education, independence, and shaped by new social relationships [15].

These psychosocial and neurodevelopmental aspects emphasize the importance of the development of certain qualities, like autonomy, self-sufficiency, intimate relationships, or educational engagement [8,11]. The challenging aspect of this stage is that youth act as scriptwriters of their own pathway, relying on their own resources, qualities, and abilities, when facing their contexts [16]. In order to understand the multiple factors that influence young adults we must distinguish between domains [8,17] and promote the elementary supports they need in order to reach some degree of cognitive and psychological maturity, as well as some degree of resilience.

### 1.2. Psychological Outcomes of Emotional Well-Being in Young Adults

#### 1.2.1. The Competency Model for Studying, Learning and Performing under Stress (SLPS)

The transition to college is rife with stressors for young adults. Their positive and negative experiences during this period affect their outcomes, being their academic performance one of the most important. Negative emotions such as stress can emerge with renewed intensity, influencing their emotional, motivational, and affective state, thus conditioning the learning process. Several studies have pointed out that stress interferes with the cognitive and motivational processes with respect to study and learning [18,19,20,21].

To prevent some of the negative consequences of stress, a preventive model should be developed, one which evaluates whether students come equipped with some tools that helps them manage stress. The SLPS model [18,22] is a competencies model designed to understand the different levels of learning that take place in stressful contexts. This model is based on the 3P model (Presage, Process, and Product) [23]. Presage variables are previous and stable traits of the student. Process variables are a set of competences that allows the student to face stressful situations. Finally, product variables refer to quantitative, qualitative, and affective learning outcomes [24]. According to this model, the learning process of the student is affected by different characteristics that need to be assessed as elements that mediate their learning. Some of the variables involved in this process are described below.

#### 1.2.2. Coping Strategies as a Meta-Emotional Variable

Coping has been defined as the “cognitive and behavioral efforts to manage specific external and/or internal demands that are appraised as taxing or exceeding the resources of the person” [25]. Two main types of coping strategies have been described: problem-focused and emotion-focused strategies. According to the American Psychological Association (APA), problem-focused strategies (e.g., confrontative coping, conflict resolution, goal setting, and support seeking) aim to reduce or eliminate the stressor or the environmental causes of stress. Emotion-focused strategies (e.g., distracting, positive reappraisal, compensatory behaviors, and emotional disclosure) regulate the negative emotional responses associated with the stressor. Instead of changing the stressor itself, cognitive and behavioral strategies are used to reduce the emotion. Both strategies are complementary in the process of facing stress. The preference and utility of each one will depend on the characteristics of the stressor, the context in which it develops, the outcome itself, and the individual resources that the person has [26].

The stability of coping strategies has been widely debated. While some authors assert that coping is a continuously dynamic and flexible process that varies according to life changes [26,27], others consider them stable with a trait-like nature [28]. The study of coping has implications on the way young people deal with stress. Recently, it has been described as a meta-ability [29] because it has behavioral implications in relation to the skills needed when facing stressors. This has implications in the learning process and in the transitions involved in college which might generate certain degree of stress. Coping is fundamental because it is involved with learning processes. For this reason, coping strategies are related with other factors such as resilience or self-regulation, which protect against negative well-being outcomes [30].

#### 1.2.3. Resilience as Meta-Motivational Variable

Resilience has positive effects on the well-being and developmental process of young adults [31]. From a developmental point of view, there are two ways of understanding resilience: as a process or as an outcome. For the purpose of this paper, resilience is understood as an “outcome of successful adaptation to adversity” [32].

It is considered as a meta-motivational ability [24] because the successful adaptation of adversity will influence not only their academic sphere, but also their social and personal development. It has been previously described as a predictor of effort, achievement motivation and positive emotions [33]. In order for young adults to develop competencies and be mentally strong, they should develop resilience [34]. Specifically, in the academic context, resilience is involved in the process of learning and adapting to college. It helps regulate the student’s motivation to act and face stressors, since the learning process is a process that requires effort and consistency [33]. Because of its wide scope, interest has been generated in adopting a lifespan perspective determining the resilience factors that influence this period of development [35]. Resilient youth are those who have faced significant adversity and adapt well to expectations in different psychosocial domains [31,36,37]. Individual resources play an important role in the successful adaptation to the new challenges they face during their transition to adult life. Coping skills, self-efficacy, goal establishment, self-control, responsibility, self-concept, and self-regulation are some of the resources that promote resilience [30,35,38,39].

However, most of the studies related to resilience have three limitations. First, most studies have been conducted in children, adolescents, and adults as a whole. Little is known about resilience in young adults. Second, the few studies available on this sample are focused on the factors that promote resilience, not on the developmental aspect of it, resulting in multiple ways of assessing resilience. Third, because resilience is a variable related to adversity, samples are usually non-representative of youth in general (e.g., people who suffered abuse, risk factors, poor mental health, low income, etc.).

#### 1.2.4. Self-Regulation as a Meta-Behavior Variable

Self-regulation has drawn a lot of attention in recent decades. Multiple studies have resulted in numerous ways of defining and measuring it. Following Zimmerman (2010), we refer to self-regulation as the planning of thoughts, emotions, and actions for the acquisition of goals [40]. It has been considered as a meta-skill because it encompasses and regulates other affective, cognitive, and behavioral variables [29,41]. Due to its wide scope, it is frequently confused with other constructs such as self-control, self-efficacy, or autonomy [42].

In the academic field, self-regulation allows an adequate performance that facilitates the development of competency and autonomy [29] and helps in coping with stress [43]. Self-regulation has been associated with positive outcomes that predict well-being and health. In relation to stress, it has been demonstrated that people with higher levels of self-regulation have diminished levels of stress [44] and higher resilience to overcome the adversities of daily life [30]. It has been described as a predictor of flourishing and mental health and shown to be negatively correlated with maladaptive behaviors such as procrastination [6]. It helps in dealing with the stressors of daily life and contributes to the development of individual competencies.

#### 1.2.5. Positivity as an Attitudinal Variable

Another protective factor that has gained increased attention in Positive Psychology is positivity. In the last few years, there has been a keen interest in understanding the competencies and personal abilities involved in the optimal performance and well-being of the individual [45,46]. Not to be confused with positive emotion or positive affect, positivity relates to the way in which we perceive the world around us; by stressing the positive, it facilitates a more effective coping in grief and adversity [47].

Transitional periods are associated with new contexts, stressors, tasks. Positivity helps young people to effectively deal with internal affective states and to actively cope with changing and demanding environments [48]. It has been defined as “the hallmark” and most significant predictor of subjective well-being because it affects how life is perceived and judged [49,50]. It has also been shown to buffer against negative emotional states and to promote resilience against physical and mental illness [51].

### 1.3. Developmental Gender and Age Differences in Young Adults 

#### 1.3.1. Gender Differences

Gender might explain differences in the presence of psychosocial competencies through adolescence and emerging adulthood. Multiple studies show that women exhibit greater abilities of emotional intelligence [52,53,54]. However, women are also more vulnerable college settings, obtaining higher levels of stress, anxiety, and depression than males [55]. Women tend to regulate their stress through social support strategies, while men use more planning and positive reevaluation when coping with stress [56]. A study revealed that men obtained higher results in active coping strategies while women were more inclined to use avoidance oriented strategies—like distraction and behavioral disengagement [57]. On the contrary, men tend to “forget about it” in order to regain control, while women “talk about it” when dealing with stress [58]. Other studies found no significant differences in coping strategies across gender [1,59,60]. Different explanations have been given to these results. First, it might be due to the fact that college students are more liberal, expecting more egalitarian gender roles [59]. Second, as they enter adulthood, the need to be part of a group might decrease, equalizing demands and stressors between males and females [26]. Their levels of stress and cognitive flexibility become more comparable.

Resilience correlates directly with gender and age [61]. Gender plays a crucial role in the development of resilience and the building of adaptation pathways [62]. Women report higher levels of resilience than men, but some males catch up as they transition into adulthood [63,64]. González-Arratia et al. found that women obtained higher scores in external factors, while men scored higher in internal factors, meaning that women require social support in order to be resilient [65]. However, a contrasting result is given by Fínez and Moran establishes that men have higher resilience than women, showing more capacity to face the obstacles typical of the academic environment [66].

Gender, however, was not significantly related to the self-regulation [67,68,69]. This might be explained by the fact that self-regulation skills are adaptable to the circumstances, and young men and women face similar stressors proper of their age [70]. However, differences were found in other components related to positivity [71,72,73] such as optimism—men score higher than women [74,75]—and positive affect—higher in women because they tend to express more their feelings with others [76,77,78,79,80].

#### 1.3.2. Age Differences

Most of the literature about age differences in developmental coping strategies and psychological adjustment is based on an adolescent sample, not on university students. However, we expect major differences in psychological maturity between adolescents, young adults, and adults. Changes in the primary sources of stress, neurobiological development, and cognitive flexibility all suggest that age contributes to the development of coping strategies [60]. Recent studies suggest that, during young adulthood, the use of avoidance strategies decreases, as support seeking and engagement strategies become more common [81,82]. A longitudinal study of young adults found that coping styles evolve through late adolescence and early adulthood, as individuals increasingly adopt problem-oriented strategies and resort less often to maladaptive emotion-focused strategies [83]; other studies also document an increase, with adulthood, of problem-oriented strategies and a decrease in avoidant strategies [60]. The decline in emotion-oriented strategies might be related to the perception that stressors are more controllable because of individual gains in autonomy over circumstances.

College students with higher levels of resilience had higher levels of perceived social support from their friends and families [84]. Longitudinal studies of young adults [85,86] confirm the importance of the relational context—family and friend—as a critical factor in resilient youth [35]. It predicts a positive adaptation to college and to other transitions appropriate of this age. From a developmental view, this age is a window of opportunity to positive change, especially among the more disadvantaged [31,36,87]. In a comparative study, resilience in young individuals was related to social support, while among the older adults it was associated with emotion regulation and problem-solving [88].

In the vulnerable period of the transition to adulthood, goal achievement and healthy behaviors require major cognitive and behavioral efforts [89]. Self-regulation helps young people be more tolerant of the adversities and failures of young adulthood [17]. It plays a protective role from exploration practices such as alcohol consumption [90], risky sexual practices [91], and other compensatory behaviors typical of this age. Together with self-control, self-regulation predicts positively personal, emotional, and cognitive well-being [92]. Circumstances are also part of the process of acquiring self-regulation: there is an interaction between the individual and the context [17,93].

Finally, positivity proves crucial for young adults as the ability that helps them cope with the new demands and challenges arising from new roles and circumstances (university, a new job, or the exploration of romantic relationships) [45]. Positivity can be promoted, despite the familiar adversities, by positive parenting [94]. Carstensen et al. studied differences in positivity between young and adults, arguing that adults tend to focus more on the positive aspects and on the stability of their well-being [95]. With respect to goal setting, the perception of an uncertain and distant future, typical in youth, may bias their outlook, making them fall more easily on negativity [96].

#### 1.3.3. Aims and Hypotheses

The diversity of new stressors that college students face makes this critical period for the study of variables involved in positive well-being. An improved understanding of the developmental factors that contribute to well-being will enlighten future interventions aimed at improving young people’s resources to face adversity. Therefore, this paper has three objectives: (1) to describe the linear association between gender, age, and emotional well-being (coping strategies, resilience, self-regulation, and positivity) in young college students; (2) to establish a linear prediction model predicting emotional well-being; (3) to investigate the presence of interaction effects between gender and age groups in the level of emotional well-being in youth. To this end, the following hypothesis are established:(1)Age and gender will be positively associated with emotional well-being. Particularly, females will correlate positively with coping strategies and self-regulation; age will be positively associated with self-regulation and resilience.(2)Given developmental differences, gender will have a better predictive power with respect to emotional well-being than age.(3)An interaction effect is expected between age and gender, and the emotional well-being variables. Particularly, the interaction between age group and gender will predict better the meta-behavioral variable (self-regulation) and the meta-emotional variable (coping strategies).

## 2. Method

### 2.1. Participants

The sample consisted in 1310 college students from 17 to 25 years old ($\bar{x}$ = 19.9, SD = 1.8). 25% of the participants were men (*n* = 327) and 75% were women (*n* = 983). Students were enrolled in Psychology, Primary Education, and Educational Psychology programs. Participants were from different universities in Spain.

### 2.2. Instruments

Coping strategies: Short Spanish version of the Coping Strategies of Stress Scale (EEC-short) [97]. The original instrument [98] consists of a 90-item self-report scale that evaluates 7 basic coping styles: problem-solving coping, negative self-focused coping, positive reappraisal, overt emotional expression, avoidance, social support seeking, and religious coping. The EEC-Short version consists of a 64-item scale with two dimensions and 10 factors for each dimension (see Table 1). The two dimensions evaluated are (D1) emotion-focused coping, and (D2) problem-focused coping. Validity measures are correct (χ2 = 478.750; DF = 58; *p* < 0.001; χ^2^/DF = 8.24; RMSM = 0.08; NFI = 0.901; RFI = 0.945; IFI = 0.903, TLI = 0.951, CFI = 0.903, RMSEA= 0.07; α = 0.93).

Resilience: Spanish version of the Connor Davison Resilience Scale (CD-RISC) [30]. The original scale [99] consists of a 25-item self-report scale assessing the ability to cope with adversity. This scale has 5 factors: (1) personal competence, high standards and tenacity, (2) self-confidence, tolerance of negative affect and strengthening effects of stress, (3) positive acceptance of change, and secure relationships, (4) control, and (5) spiritual influences. The Spanish version was first translated by [100] and later validated by [101], resulting in a 17-item scale with three factors: tenacity, personal control, and social competence (α = 0.79, χ^2^ = 198, df = 116, *p* < 0.001; χ^2^/DF = 1.70; RMSM = 0.041; CFI = 0.902; SRMR = 0.062). Moreover, [102] validated the Spanish version of the 10-item CD-RISC with a one-factor model (α = 0.88, χ^2^ = 68.215, df = 35, *p* = 0.001; CFI = 0.96). This one-factor model has been backed up by [103], with a Spanish sample representative of the general population. The CD-RISC version consists of a 25-item scale with 5 factors: (1) tenacity and personal competence, (2) stress tolerance, (3) perception of control and achievement, (4) perception of support, and (5) tolerance of negative situations (see Table 2). The validity measures were appropriate (α = 0.751, χ^2^ = 100,856; *p* < 0.05; SRMR = 0.052; RFI = 0.921; IFI = 0.968, TLI = 0.957, CFI = 0.963, RMSEA= 0.027).

Self-regulation: Spanish Short Self-Regulation Questionnaire (SSSRQ) [104]. The original version designed by Brown, Miller and Lawendowski (1999) consisted of a 63-item self-report scale assessing 7 dimensions of self-regulation: informational input, self-monitoring progress, motivation for change, work, and re-evaluation of the plan. Pichardo et al. (2014) studied the factor structure and internal consistency of the original SRQ, extracting a shorter version for a Spanish sample, resulting in a 17-item scale with four factors: goal setting, perseverance, decision making, and learning from mistakes [105]. Finally, Umerenkova et al. (2017) validated an abbreviated version of the Spanish questionnaire with a Rasch Analysis, resulting in a 17-item, 5-point Likert Scale [6] (see Table 3). The validity measures were appropriate (α = 0.86; χ^2^ = 641,209; *p* < 0.001; RFI = 0.949; IFI = 0.972, TLI = 0.994, CFI = 0.992, RMSEA = 0.075).

Positivity: Positivity Scale (PS). It was originally designed by Caprara et al. (2012) and consists of a 10-item self-report scale that asks participants statements related to positivity like self-esteem (e.g., “I usually have a lot of confidence in myself”), optimism (e.g., “I look to the future with hope and enthusiasm”), and life satisfaction (e.g., “I am satisfied with my life”) [106]. Item response options range from 1 (totally disagree) to 5 (totally agree). The Spanish validation showed appropriate validity measures (χ^2^ = 308.992; *p* < 0.001; NFI = 0.901; RFI = 0.894; IFI = 0.912 TLI = 0.923, CFI = 0.916; RMSEA = 0.085; HOELTER = 260 (*p* < 0.05) and 291 (*p* < 0.01); α = 0.893; Part 1 = 0.832, Part 2 = 0.813; Spearman-Brown = 0.862; Guttman = 0.832).

Gender and Age: Gender was assessed as a dichotomic variable (1 = men; 2 = women) and age ranged from 17 to 25 years old. Three homogeneous groups were created: group 1 “younger” (17–19 years old), group 2 “middle” (20–22 years old), and group 3 “older” (23–25 years old).

### 2.3. Procedure

Data were collected through an e-utility (a tool designed to be an online self-help application that offers guidance to students as they make choices in stressful contexts), in the context of two R&D Projects (2018–2021; see Funding). Participants were recruited by convenience from three public universities in Spain. The study was explained to the teachers of the corresponding degrees, and those who accepted to participate were the ones that distributed the survey to their students. Students were invited to participate through an online application (www.inetas.net). Informed consent was obtained before they completed the survey, and the aims of the study were clarified. None of the participants were compensated financially nor academically for taking the survey. The data collected is anonymous and is protected by the actual Spanish Legislation and the Code Deontology of the Official College of Psychology of Spain. This study has been approved by the Ethics Committee and the corresponding Institutional Review Boards (ref. 2018.170).

### 2.4. Data Analysis

A cross-sectional study design has been carried out. Using an ex-post facto data analysis design, descriptive and inferential analyses have been conducted [107]. To test Hypothesis 1, an association analysis was executed through a bivariate two-tail Pearson correlation to assess the relationship between gender, age, and the emotional well-being variables. For Hypothesis 2, a linear and multiple prediction analysis was carried out to explore the effect that age and gender have on emotional well-being. Finally, for hypothesis 3, an inferential multivariate analysis of variance was applied, to evaluate the interaction effect between gender and age, as predicting variables, and the emotional well-being variables as criterion variables. Further, the effect of gender and age on the emotional well-being variables was assessed. The statistical program used was IBM SPSS (version 22, IBM, Armonk, NY, USA).

## 3. Results

### 3.1. Association Effects

#### 3.1.1. Coping Strategies

A positive relation was found between gender and the total score of coping strategies. There was no association between gender and emotion-focused strategies (D1). However, a positive and significant relationship was found with problem-focused strategies (D2). Specifically, with three factors: seeking family help and counsel (EECF2), self-talk (EECF5), and communicating feelings and social support (EECF12). A negative correlation was found between gender and with resigned acceptance (EECF11).

Age did not correlate with the total score of coping strategies or any of its dimensions, but it did with some of its factors. Regarding emotion-focused strategies (D1), a negative association was found with reducing anxiety and avoidance (EECF7). Negative correlations were found with some problem-focused strategies (D2) such as seeking family help and counsel (EECF2) and communicating feelings and social support (EECF12), and a positive association with positive reappraisal (EECF10). See Table 4.

#### 3.1.2. Resilience

Gender is positively correlated with the total score for resilience and some of its factors such as control perception. A negative significant association was found with stress tolerance. Age, however, was positively associated with tenacity and stress tolerance. Interestingly, spirituality strongly correlates both with gender (positively) and age (negatively). See Table 4.

#### 3.1.3. Self-Regulation

A positive and significant association was found between the total score of self-regulation and age. More specifically, this is true for self-regulation factors such as perseverance and decision making. Gender only correlated positively with goal establishment. See Table 4.

#### 3.1.4. Positivity

Neither age nor gender correlated significantly with positivity. See Table 4.

### 3.2. Prediction Effects

#### 3.2.1. Coping Strategies

The linear regression showed that gender is the variable that best predicts coping, especially the dimension of problem-focused coping (D2) and some of its factors such as help seeking (EECF2) and resigned acceptance (EECF11). Communicating feelings and social support (EECF12) was negatively predicted by gender. Age did not predict for any of the dimensions of coping strategies, but it did for some of its factors; negatively with anxiety reduction and avoidance (EECF7) and help seeking (EECF2), and positively with positive reappraisal (EECF10).

#### 3.2.2. Resilience

In relation with resilience, gender predicted some of the resilience factors such as stress tolerance, control perception, and spirituality. Age, however, predicts all the resilience factors except tolerance to change. The spirituality factor was strongly predicted positively by gender and negatively by age.

#### 3.2.3. Self-Regulation

Goal establishment was the only self-regulation factor predicted by gender (see Table 5). Age is the variable that best significantly predicts the total score of self-regulation, along with two of its factors, perseverance and decision making.

#### 3.2.4. Positivity

Neither age nor gender correlated significantly with positivity. See Table 5.

### 3.3. Inferential Effects

#### 3.3.1. Coping Strategies

Multivariate analysis with gender [2 levels; male-female] and age groups (3 levels; younger (17–19 years old), middle (20–22 years old), older (23–25 years old)) as independent variables show that gender has a high explanatory power for coping strategies. It determines the differences of coping more specifically with the dimension of problem-focused strategies (D2) and some of its factors such as help seeking (EECF2), self-talk (EECF5), positive reappraisal (EECF10), communicating feelings and social support (EECF12), and seeking alternative reinforcement (EECF13). In relation with the emotion-focused strategies (D1) gender only had an effect with resigned acceptance (EECF11).

Age does not determine changes in any of the dependent variables. However, has an explanatory power in four coping factors. It has a slight but significant explanatory power with help seeking (EECF2), anxiety reduction and avoidance (EECF7) and with communicating feelings and social support (EECF12). An interaction effect was found with problem-focused strategies, and specifically with positive reappraisal and firmness (EECF10). See Table 6.

#### 3.3.2. Resilience

A main effect was found between gender and resilience, especially in control perception and spirituality. Age has an effect on tenacity and spirituality. Interestingly, spirituality was significantly predicted by both gender and age, but no interaction effect was found. Interaction effects were found with stress tolerance and change adaptation factors. See Figure 1 and Figure 2.

#### 3.3.3. Self-Regulation

Gender was the only independent variable that strongly predicted self-regulation. A prediction effect is found with goal establishment and perseverance. As for age, the only factor that had an effect with it was perseverance. An interaction effect was found for goal establishment. See Table 6.

#### 3.3.4. Positivity

An interaction effect was found for the total scores of positivity. No inferential effects were found with age or gender separately. See Table 6.

## 4. Discussion

**Hypothesis** **1.***Age and gender are expected to be positive associated with emotional well-being variables*.

As predicted, positive associations were found between gender, age, and the well-being outcomes. Gender had a strong positive association with coping strategies, especially those that are problem-focused, and with self-regulation. According to our results, seeking for help, actions directed to causes, self-instructions, and communicating feelings and seeking for social support are coping strategies associated positively with women. These results are contrary to previous studies [57] but they might reflect stable cultural differences because our results are consistent with prior studies with a Spanish population [56]. Negative associations are interesting as well. Gender was negatively correlated with resigned acceptance (coping strategies factor) and stress tolerance (resilience factor). This might be explained by women being more vulnerable to stressful events and to tend to experience higher levels of it than males [55]. The resilient factors of control perception and spirituality were positively associated with gender. This is contrary to mention of men being higher in internal resilient factors and women requiring external social support to be resilient [65]. However, our results also show that spirituality is a factor that is negatively related with age, meaning that even women tend to be higher in this aspect, this association decreases over the course of years. Perseverance, decision making, help for action, positive reappraisal and tenacity are characteristics that are positively associated with age. As several authors point out [60,81,83] the use of negative strategies become more stable as they mature. This might be related to the multiple stressors they have been exposed through time, gaining more control on their circumstances.

**Hypothesis** **2.***Gender will have a better predictive power with emotional well-being variables*.

As noted, gender was the dependent variable that had a better predictive strength regarding coping strategies. More specifically with problem-focused strategies, help seeking, actions directed to causes, and feelings communication and social support. Spirituality was the only resilient factor that was highly predicted by gender. For the other well-being outcomes, gender did not have predictive power. However, age had higher predictive power with self-regulation, more specifically with the perseverance factor. Some self-regulation factors were predicted by age such as religious support, anxiety reduction, and positive reappraisal, as well as the resilient factors of tenacity and spirituality. It makes sense that self-regulation is highly predicted by age, because it has been previously noted as a meta-skill that helps regulate other variables [29,41] and is helpful when dealing with the stressors of daily life [108]. As a meta-behavioral variable, it correlates with other important competencies such as autonomy [42], which are crucial for the other well-being variables that were predicted, such as resilience [30]. These results are of relevance to the conceptualization of self-regulation as a meta-skill because it relates to other protective factors of well-being [33]. However, it still not clear if these are only dependent on the passing of the years, or if other contextual variables influence this process, for example, attending college or if one attends a public or private university. More information is needed to clarify these findings, but it is a first approach to start conceiving self-regulation as a competence associated with other protective variables of well-being.

**Hypothesis** **3.***Interdependence and interaction effects are expected between age group and gender, and the emotional well-being*.

It was predicted that the interaction between age group and gender will predict better the meta-behavioral variable (self-regulation) and the meta-emotional variable (coping strategies). According to our results, the interaction between the independent variables did not predict any of the meta-skills. Consistent with previous discussion, it seems that gender is the best predictive variable for coping strategies, resilience, and self-regulation factors. The interaction between age and gender only predicted resigned acceptance, positive reappraisal, tolerance to change, goal establishment, and total positivity. The explanation might be that gender is a more solid predictor because sex differences might later explain differences in developmental psychosocial variables [52]. There are few studies that undermine the importance of gender-based differences. Despite the fact that gender roles have become more equitable through years [26], these outcomes show that gender itself is a variable that is involved in variables that are crucial for youth development. These results do not imply that women are better than men. What it is noted here is that the developmental processes of males and females are different [65], and the way they cope in college -one of the most important and stressful period in life- will be different.

### Limitations

This investigation has several limitations that ought to be noted. First, the sample is not randomized nor representative of Spanish youth. Most participants are college students, resulting in an under-representation of all the emerging adults that are not transitioning to college. The second limitation is that the sample is not equally distributed by gender. This might explain the important unveiled effect that gender had on the well-being outcomes. Third, since the scales are answered through an online platform, it is only available to those in the population who have access to the Internet (a very large majority in Spain, anyway). Further, psychometrically speaking, online instruments have certain disadvantages such as inattention or misunderstandings that could bias the participants’ response [109]. The dropout rate becomes easier, explaining why several questionnaires were incomplete. However, this methodology could be advantageous to access young people from different countries and obtain future transcultural comparison.

Future research should widen the sample to non-college students and recruit more males. Non-college students might not be facing academic stressors, but they are also confronted with the difficulties of the transition itself. In fact, previous studies have noted that non-college-attending students have more prevalence in developing mental health disorders, substance abuse, isolation, and low social support [110,111,112]. However, there are other chosen paths that are relevant in this population such as employment or military service that need to be addressed. As mentioned in the introduction, the traditional markers of adulthood have changed resulting in multiple ways of becoming an adult that need to be considered when studying this population. Furthermore, future research should aim at addressing the contextual factors involved in the development of well-being outcomes. As mentioned, research in Psychology should focus on the different micro, molecular and molar levels [17]. The interaction in real and various contexts is usually left aside by emphasizing the individual and personal traits. However, when talking about well-being outcomes, the context plays a primordial role. Other variables from their personal history, childhood events or other negative factors need to be assed in order to comprehend the whole picture of their developmental transition. Resilience, coping strategies, self-regulation, and positivity are influenced by other external characteristics such as education, parenting styles, core beliefs, culture, etc., which need to be addressed.

## 5. Practical Implications

### 5.1. Academic Implications

The results presented here help us turn our sight back to the importance of gender-based and age-related differences. College is one of the most stressful periods of youth. Understanding the differences between males and females in the acquisition of variables that are crucial to their well-being could help build better learning programs aimed at addressing these differences. Interventions can be directed to those students with poor academic performance. Specially, as our results prove, interventions can be designed attending to gender differences, reinforcing interventions focused on what each student needs to develop.

### 5.2. Professional Implications

The outcomes selected for this study are meta-skills that promote other competencies that are crucial for development when coping with stress in college, and also during the entire transition to adulthood. According to the CLPS theoretical model [22], meta-behavioral, meta-emotional, and meta-emotional variables should be taken into consideration, while adapting the learning process to a personalized program that addresses the characteristics of the students, like their gender and age. Therefore, we could promote an adequate psychosocial well-being in our students, one that helps them cope with future stressors and responsibilities adult life.

## 6. Conclusions

The results presented in this research stablish associative, predictive, and inferential evidence of the role of gender and age in the developmental emotional well-being outcomes of young adults. Gender and age predict coping strategies (meta-emotional variable), resilience (meta-motivational variable) and self-regulation (meta-behavioral variable). However, gender was the variable predominantly important in the acquisition of the well-being outcomes. Positivity was considered as an attitudinal variable, but it did not seem to be associated neither with gender nor age, but a slight interaction effect was found between these two. There are outcomes that do not develop automatically with age. It is commonly believed that people, as they age, will become more mature; but our results shows that the passing of the years does not produce, in itself, a change in well-being outcomes. The acquisition of some of these outcomes is not developmental. Therefore, interventions are necessary. An improved understanding of the developmental factors involved in well-being outcomes will enlighten future interventions aimed at improving young people’s resources to face adversity. These results shed a new light on the well-being literature, by proving that demographic variables, such as gender and age, are somehow linked to the developmental well-being variables. Specifically, it reconsiders the value of gender and age, and its relation to the development of certain meta-skills.

## Figures and Tables

**Figure 1 ijerph-18-00522-f001:**
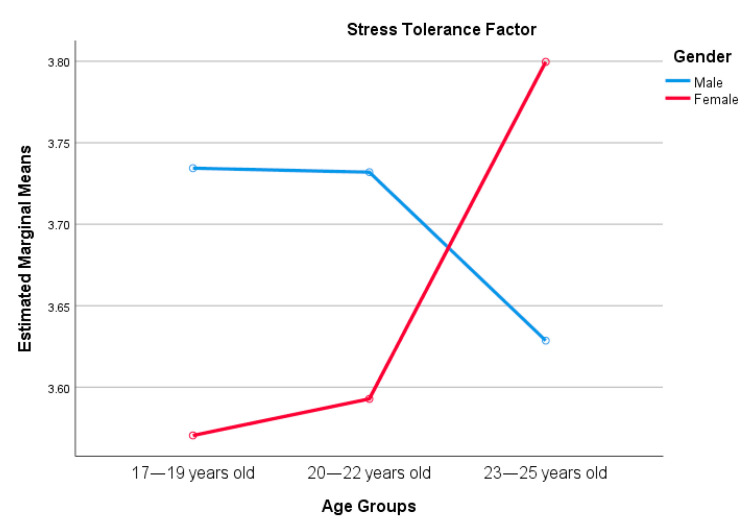
Interaction effect between gender and age, and resilience. Specifically, with the stress tolerance factor.

**Figure 2 ijerph-18-00522-f002:**
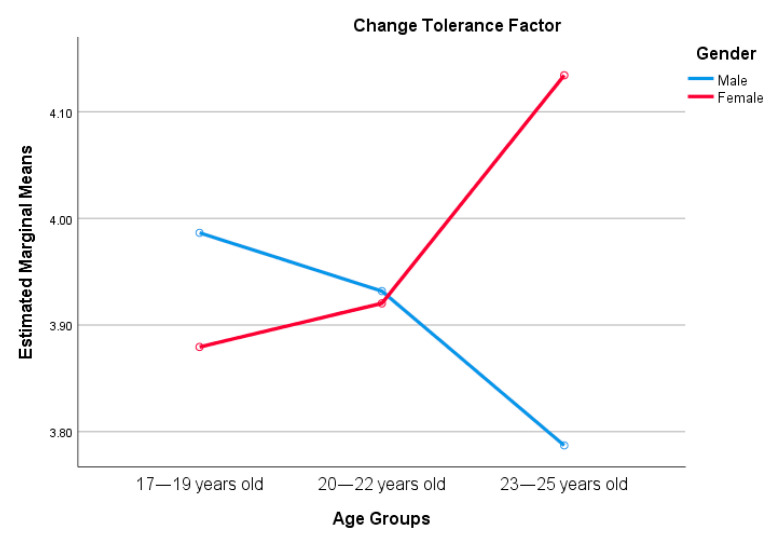
Interaction effect between gender and age, and tolerance to change.

**Table 1 ijerph-18-00522-t001:** Factors and dimensions of the Coping Strategies of Stress Scale (EEC)-short [97]. Reproduced with permission from de la Fuente, J., International Journal of Environmental Research and Public Health; published by MDPI, 2018.

**D1. Emotion-Focused Coping.**	**Example Items**
(F1) avoidant distraction	I get away and forget the problem temporarily (change of environment)
(F7) reducing anxiety and avoidance	I practice some kind of sport in order to reduce my anxiety or tension
(F8) preparing for the worst	I prepare myself for the worst
(F9) emotional venting and isolation	I act irritable and aggressive toward others
(F11) resigned acceptance	I accept the problem as it is, since I cannot do anything to solve it
**D2. Problem-Focused Coping**	**Example Items**
(F2) seeking family help and counsel	I talk with people I know who can do anything to solve my problem
(F5) self-talk	I lay out a plan of action and try to carry it out
(F10) positive reappraisal and firmness	I try to see positive aspects of the situation
(F12) communicating feelings and social support	I feel better if I explain my problem to friends or family members
(F13) seeking alternative reinforcement.	I start new activities (studies, etc.)

**Table 2 ijerph-18-00522-t002:** Factors of the Connor Davison Resilience Scale (CD-RISC) scale.

Factors	Example Items
Coping and confidence	I think I can achieve my goals even if there are obstacles
Tenacity and adaptation to change	I am able to adapt when changes arise
Perception of control and achievement	Facing difficulties can make me stronger
Perception of support	I have at least one intimate and secure relationship that helps me when I’m stressed
Tolerance to negative situations	I am able to handle unpleasant feelings

**Table 3 ijerph-18-00522-t003:** Factors of the Spanish Short Self-Regulation Questionnaire (SSSRQ) [104].

Factors	Example Items
Goal setting	I set goals for myself and keep track of my progress
Perseverance	I have a lot of willpower
Learning from mistakes	I don’t seem to learn from my mistakes
Decision making	I have trouble making up my mind about things

**Table 4 ijerph-18-00522-t004:** Bivariate Correlations between Gender and Age with Well-being Outcomes.

	Gender	Age
**Coping strategies**		
Total coping strategies	0.140 **	0.004
Emotion-focused (D1)	0.011	−0.047
EECF1	−0.006	−0.048
EECF7	0.003	−0.082 **
EECF8	0.042	−0.041
EECF9	−0.015	0.002
EECF11	−0.061 *	−0.033
Problem-focused (D2)	0.208 **	0.039
EECF2	0.185 **	−0.069 *
EECF5	0.066 *	0.056
EECF10	−0.003	0.111 **
EECF12	0.250 **	−0.058 *
EECF13	0.034	0.026
**Resilience**		
Total resilience	0.068 *	0.018
Tenacity	−0.020	0.136 **
Stress tolerance	−0.096 **	0.068 *
Change tolerance	−0.016	0.046
Control perception	0.075 *	0.059
Spirituality	0.150 **	−0.147 **
**Self-regulation**		
Total self-regulation	0.044	0.086 **
Goals	0.097 **	0.036
Perseverance	0.048	0.121 **
Decision Making	−0.044	0.072 *
Learning from mistakes	0.039	0.020
**Positivity**		
Total positivity	0.015	0.066

Note. (EECF1) avoidant distraction; (EECF2) seeking family help and counsel; (EECF5) self-talk; (EECF7) reducing anxiety and avoidance; (EECF8) preparing for the worst; (EECF9) emotional venting and isolation; (EECF10) positive reappraisal and firmness; (EECF11) resigned acceptance; (EECF12) communicating feelings and social support; (EECF13) seeking alternative reinforcement; * *p* < 0.05; ** *p* < 0.01. Bold letters: highlight the main variable and its factors.

**Table 5 ijerph-18-00522-t005:** Lineal Regression Analysis between Gender and Age, and Well-being Outcomes.

	Gender	Age	F-Value	R2
**Coping strategies**				
Total coping strategies	β = 0.141 **	β = 0.010	F(2,895)= 8.987 **	0.020
Emotion-focused (D1)	β = 0.009	β = −0.046	F(2,1035)= 1.174	0.002
EECF1	β = −0.008	β = −0.049	F(2,1196)= 1.436	0.002
EECF7	β = 0.000	β = −0.082 **	F(2,1189)= 4.035 *	0.007
EECF8	β = 0.040	β = −0.039	F(2,1170)= 1.916	0.003
EECF9	β = −0.015	β = 0.001	F(2,1169)= 0.129	0.000
EECF11	β = −0.062 *	β = −0.035	F(2,1180)= 2.935 *	0.005
Problem-focused (D2)	β = 0.209 **	β = 0.046	F(2,1027)= 24.277 **	0.045
EECF2	β = 0.186 **	β = −0.061 *	F(2,1159)= 22.799 **	0.038
EECF5	β = 0.067	β = 0.058	F(2,1171)= 4.531 *	0.008
EECF10	β = 0.001	β = 0.111 **	F(2,1170)= 7.290 *	0.012
EECF12	β = 0.248 **	β = −0.048	F(2,1202)= 41.620 **	0.065
EECF13	β = 0.035	β = 0.028	F(2,1164)= 1.119	0.002
**Resilience**				
Total resilience	β = 0.069	β = 0.022	F(2,934)= 2.376	0.005
Tenacity	β = −0.016	β = 0.136 **	F(2,996)= 9.505 **	0.019
Stress tolerance	β = −0.094 *	β = 0.065 *	F(2,986)= 6.722 *	0.013
Change tolerance	β = −0.014	β = 0.045	F(2,991)= 1.136	0.002
Control perception	β = 0.077 *	β = 0.063 *	F(2,999)= 4.762 *	0.009
Spirituality	β = 0.145 **	β = −0.142 **	F(2,1007)= 22.390 **	0.043
**Self-regulation**				
Total self-regulation	β = 0.048	β = 0.088 **	F(2,1169)= 5.683 *	0.010
Goals	β = 0.099 *	β = 0.040	F(2,1222)= 6.835 *	0.011
Perseverance	β = 0.053	β = 0.123 **	F(2,1233)= 10.927 **	0.017
Decision Making	β = −0.042	β = 0.070 *	F(2,1214)= 4.195 *	0.007
Learning from mistakes	β = 0.040	β = 0.022	F(2,1233)= 1.250	0.002
**Positivity**				
Total positivity	β = 0.019	β = 0.067	F(2,510)= 1.198	0.005

Note. (EECF1) avoidant distraction; (EECF2) seeking family help and counsel; (EECF5) self-talk; (EECF7) reducing anxiety and avoidance; (EECF8) preparing for the worst; (EECF9) emotional venting and isolation; (EECF10) positive reappraisal and firmness; (EECF11) resigned acceptance; (EECF12) communicating feelings and social support; (EECF13) seeking alternative reinforcement; * *p* < 0.05; ** *p* < 0.01. Bold letters: highlight the main variable and its factors.

**Table 6 ijerph-18-00522-t006:** Interdependence relations between age groups (AG) and gender (G) as independent variables, in well-being outcomes (dependent variables).

Age Groups (AG)	Younger (*n* = 699)		Middle (*n* = 472)		Older (*n* = 139)			Effects	
Gender (G)	Male	Female	Male	Female	Male	Female		F (Pillai Test)	Eta Squared (η^2^)
*n*	177	522	109	363	41	98			
**Coping Strategies**									
Total coping	2.65 (0.35)	2.67 (0.25)	2.52 (0.31)	2.69 (0.30)	2.68 (0.23)	2.75 (0.24)	G AG GxAG	F (1,896) = 18.652 ** F (2,896) = 0.101 F (2,896)= 1.016	0.021 0.000 0.002
Emotion-focused (D1)	2.45 (0.30)	2.46 (0.32)	2.42 (0.33)	2.42 (0.32)	2.40 (0.34)	2.41 (0.32)	G AG GxAG	F (1,1036) = 0.083 F (2,1036) = 1.669 F (2,1036) = 0.017	0.000 0.003 0.000
EECF1	2.29 (0.51)	2.31 (0.51)	2.28 (0.54)	2.25 (0.48)	2.32 (0.46)	2.19 (0.43)	G AG GxAG	F (1,1197) = 1.035 F (2,1197) = 0.690 F (2,1197) = 1.095	0.001 0.001 0.002
EECF7	3.10 (0.58)	3.10 (0.66)	3.04 (0.61)	3.01 (0.60)	2.89 (0.53)	2.96 (0.62)	G AG GxAG	F (1,1190) = 0.101 F (2,1190) = 3.998 * F (2,1190) = 0.248	0.000 0.007 0.000
EECF8	2.70 (0.40)	2.71 (0.49)	2.59 (0.46)	2.69 (0.48)	2.63 (0.49)	2.66 (0.49)	G AG GxAG	F (1,1171) = 1.552 F (2,1171) = 1.948 F (2,1171) = 0.716	0.001 0.003 0.001
EECF9	1.91 (0.48)	1.91 (0.45)	1.88 (0.46)	1.89 (0.48)	1.97 (0.40)	1.84 (0.44)	G AG GxAG	F (1,1170) = 1.350 F (2,1170) = 0.224 F (2,1170) = 0.970	0.001 0.000 0.002
EECF11	2.13 (0.52)	2.08 (0.55)	2.08 (0.48)	2.04 (0.55)	2.72 (0.55)	1.95 (0.48)	G AG GxAG	F (1,1181) = 9.505 * F (2,1181) = 0.885 F (2,1181) = 3.069 *	0.008 0.002 0.005
Problem-focused (D2)	2.81 (0.35)	2.94 (0.33)	2.79 (0.36)	2.96 (0.31)	2.72 (0.29)	3.05 (0.35)	G AG GxAG	F (1,1028) = 50.947 ** F (2,1028) = 0.068 F (2,1028) = 3.331 *	0.047 0.000 0.006
EECF2	2.82 (0.71)	3.03 (0.65)	2.67 (0.75)	3.00 (0.65)	2.50 (0.68)	3.03 (0.63)	G AG GxAG	F (1,110) = 41.259 ** F (2,1160) =3.065 * F (2,1160) = 2.288	0.035 0.005 0.004
EECF5	3.01 (0.43)	3.05 (0.43)	3.01 (0.44)	3.07 (0.43)	2.93 (0.44)	3.19 (0.42)	G AG GxAG	F (1,1172) = 10.483 * F (2,1172) = 0.198 F (2,1172) = 2.779	0.009 0.000 0.005
EECF10	3.08 (0.45)	2.99 (0.48)	3.04 (0.52)	3.09 (0.46)	2.99 (0.47)	3.26 (0.49)	G AG GxAG	F (1,1171) = 3.784 * F (2,1171) = 1.654 F (2,1171) = 6.702 *	0.003 0.003 0.011
EECF12	2.82 (0.77)	3.15 (0.70)	2.68 (0.77)	3.18 (0.66)	2.48 (0.64)	3.11 (0.72)	G AG GxAG	F (1,1203) = 71.742 ** F (2,1203) = 3.417 * F (2,1203) = 2.637	0.057 0.006 0.004
EECF13	2.85 (0.44)	2.86 (0.43)	2.81 (0.42)	2.85 (0.41)	2.81 (0.37)	2.97 (0.44)	G AG GxAG	F (1,1165) = 3.681 * F (2,1165) = 0.712 F (2,1165) = 1.258	0.057 0.006 0.004
**Resilience**									
Total resilience	3.65 (0.54)	3.59 (0.51)	3.60 (0.53)	3.69 (0.45)	3.60 (0.58)	3.95 (0.39)	G AG GxAG	F (1,935) = 8.779 * F (2,935) = 1.187 F (2,935) = 2.301	0.009 0.003 0.005
Tenacity	3.92 (0.60)	3.82 (0.57)	3.91 (0.62)	3.92 (0.57)	3.94 (0.71)	4.14 (0.50)	G AG GxAG	F (1,997) = 0.582 F (2,997) = 3.022 * F (2,997) = 2.56	0.001 0.006 0.005
Stress	3.73 (0.56)	3.57 (0.51)	3.73 (0.57)	3.59 (0.54)	3.62 (0.51)	3.79 (0.53)	G AG GxAG	F (1,987) = 0.856 F (2,987) = 0.475 F (2,987) = 3.582 *	0.001 0.001 0.007
Change	3.98 (0.61)	3.87 (0.60)	3.93 (0.68)	3.92 (0.58)	3.78 (0.64)	4.13 (0.51)	G AG GxAG	F (1,992) = 2.018 F (2,992) = 0.109 F (2,992) = 5.089 *	0.002 0.000 0.010
Control	3.83 (0.79)	3.87 (0.71)	3.83 (0.78)	4.01 (0.69)	3.74 (0.82)	4.12 (0.64)	G AG GxAG	F (1,1000) = 9.801 * F (2,1000) = 0.866 F (2,1000) = 2.335	0.010 0.002 0.005
Spirituality	3.19 (1.01)	3.56 (0.96)	2.91 (1.04)	3.22 (0.93)	3.06 (1.02)	3.32 (0.87)	G AG GxAG	F (1,1008) = 13.271 ** F (2,1008) = 8.572 ** F (2,1008) = 0.165	0.013 0.017 0.000
**Self-regulation**									
Total self-regulation	3.56 (0.44)	3.42 (0.59)	3.33 (0.56)	3.55 (0.60)	3.39 (1.01)	3.76 (0.44)	G AG GxAG	F (1,1170) = 5.925 * F (2,1170) = 1.250 F (2,1170) = 2.117	0.005 0.002 0.004
Goals	3.71 (0.66)	3.76 (0.63)	3.61 (0.70)	3.83 (0.64)	3.51 (0.75)	3.88 (0.55)	G AG GxAG	F (1,1223) = 17.130 ** F (2,1223184 F (2,1223) = 3.497 *	0.014 0.000 0.006
Perseverance	3.21 (0.82)	3.21 (0.79)	3.22 (0.77)	3.41 (0.88)	3.28 (0.77)	3.51 (0.75)	G AG GxAG	F (1,1234) = 4.779 ** F (2,1234) = 3.169 * F (2,1234) = 1.651	0.004 0.005 0.003
Decision making	3.32 (0.73)	3.20 (0.76)	3.38 (0.69)	3.26 (0.79)	3.26 (0.81)	3.47 (0.68)	G AG GxAG	F (1,1215) = 0.007 F (2,1215) = 1.195 F (2,1215) = 2.267	0.000 0.002 0.004
Learning from mistakes	3.50 (0.87)	3.53 (0.91)	3.44 (0.99)	3.56 (0.94)	3.46 (0.97)	3.71 (0.80)	G AG GxAG	F (1,1234) = 3.251 F (2,1234) = 0.330 F (2,1234) = 0.682	0.003 0.001 0.001
**Positivity**									
Total positivity	3.69 (0.75)	3.73 (0.64)	3.89 (0.51)	3.80 (0.56)	3.53 (0.63)	3.91 (0.61)	G AG GxAG	F (1,511) =2.068 F (2,511) = 2.793 F (2,511)= 4.376 *	0.004 0.011 0.017

Note 1. (EECF1) avoidant distraction; (EECF2) seeking family help and counsel; (EECF5) self-talk; (EECF7) reducing anxiety and avoidance; (EECF8) preparing for the worst; (EECF9) emotional venting and isolation; (EECF10) positive reappraisal and firmness; (EECF11) resigned acceptance; (EECF12) communicating feelings and social support; (EECF13) seeking alternative reinforcement; G = gender; AG = age group; GxAG = interaction between gender and age group; * *p* < 0.05; ** *p* < 0.01. Bold letters: highlight the main variable and its factors.

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
