# Peer review of "The Role of Gender and Age in the Emotional Well-Being Outcomes of Young Adults"

_ijerph, 2021, doi:10.3390/ijerph18020522_

Round 1
Reviewer 1 Report
Minor edits suggested in discussion and conclusions sections. Please see attached.
The Role of Gender and Age in the Emotional Well-being Outcomes of Young Adults is a clearly written and well-organized research manuscript regarding the emotional well-being of young adults. This is an important topic for complex time-period in life as teenagers become adults and it is also a timely topic during this global pandemic that is impacting not only physical but also mental health. The study examines the role of gender and age in how youth, who are transiting to college, cope with stress. It was hypothesized that age and gender are positively associated with emotional well-being, gender is a better predictor than age, and there is an interaction effect between age and gender.
The literature review is extensive, detailed, and uses current literature, thus providing a solid foundation on which the study is positioned and the results are interpreted. The quantitative research methods are appropriate to the study and use suitable instruments such as the Spanish Short Self-Regulation Questionnaire and the Spanish version of the Connor Davison Resilience Scale.
The results of the research were clearly accounted for each of the three hypotheses, including where the data did not support the hypothesis. The manuscript makes good use of tables and figures throughout to facilitate understanding.
Questions of the representation of the sample and participant access to the online platform are addressed in the section on limitations.
There are some challenges in the manuscript. The first are minor grammatical edits. (See the submitted suggestions done via track changes.) The second challenge is the focus of the study on college students. This could simply be more clearly noted in the limitations section of the manuscript and perhaps that future studies might include other teen-to-adult pathways. Both of these challenges can be addressed with minor edits.
Overall this is an important study that focuses on a shifting period of life and, at a time in current history when mental health and well-being are timely topics worldwide. The manuscript is well written, is founded on existing research literature, and contributes to the knowledge of youth well-being.

Reviewer 2 Report
Thank you for the opportunity to review the manuscript "The Role of Gender and Age in the Emotional Well-being Outcomes of Young Adults." This was a very interesting study to read, and I believe it makes some important contributions to the field. I have outlined several comments and questions below, mostly organized by section.
Intro-
1. When discussion the five main domains, perhaps some clarification about what I'm assuming is the fourth stage "in-between." As it currently reads, it seems more tied to the third stage that is mentioned.
2. While there is a focus on young adults transitioning to college and how this may impact their psychological and developmental outcomes, there should be some mention of individuals who choose to not go to college. For example, individuals who may enter military service, enter employment, or some other career/education combination that is not the traditional college or higher education.
3. You state rather clearly how you're considering resilience as an outcome for the purposes of this paper, but I'm wondering if there's any utility in also looking at is as a process. This period of time for young adults is clearly a time of change and adaptation. I wonder how thinking about resilience as also a process rather than simply an outcome may also factor into this model.
Methods-
4. Was there additional reasoning behind recruiting from the specific programs? These programs tend to have more females than males, so that may explain the differences in gender for the overall sample characteristics.
5. As gender is a dichotomous variable and age coded as a categorical variable, using Pearson's correlation wouldn't be the accurate test to use. Did you consider using independent samples t-test or ANOVA to look at the relationship between these variables with emotional well-being?
Discussion-
6. Were other variables collected during the recruitment process that may relate to some of the negative or non-expected findings, specifically potential confounding variables related to coping or resilience? For these, I'm thinking adverse childhood events (ACEs) that may make an individual more susceptible to possible negative coping strategies or instead could result in posttraumatic growth and increased resilience.
Round 2
Reviewer 2 Report
Thank you for your thoughtful responses to my initial concerns. I am happy with the manuscript in its current form and do not have any additional comments or questions.